# Cerebrospinal fluid levels of neuron-specific enolase predict the severity of brain damage in newborns with neonatal hypoxic-ischemic encephalopathy treated with hypothermia

Marisol-Zulema León-Lozano[1,2], Juan Arnaez[3,4], Ana Valls[5], Gemma Arca[4,6], Thais Agut[4,7], Ana Alarcón[7], Alfredo Garcia-Alix[2,4,5,8]*

1 Althaia Xarxa, Assistencial Universitária de Manresa, Barcelona, Spain, 2 University of Barcelona, Barcelona, Spain, 3 Department of Neonatology, Hospital Universitario de Burgos, Burgos, Spain, 4 NeNe Foundation, Madrid, Spain, 5 Institut de Recerca Sant Joan de Dèu, Hospital Sant Joan de Déu, Barcelona, Spain, 6 Department of Neonatology, Hospital Clinic, IDIBAPS, Barcelona, Spain, 7 Department of Neonatology, Hospital Sant Joan de Déu, Barcelona, Spain, 8 CIBER de Enfermedades Raras, Madrid, Spain

* alfredoalix@gmail.com

**Data Availability Statement:** All relevant data are within the manuscript and its Supporting Information files.

## Abstract

### Objectives

To investigate whether cerebrospinal fluid levels of neuron-specific enolase (CSF-NSE) during the first 72 hours correlate with other tools used to assess ongoing brain damage, including clinical grading of hypoxic-ischemic encephalopathy (HIE), abnormal patterns in amplitude integrated electroencephalography (aEEG), and magnetic resonance imaging (MRI), as well as with the neurodevelopmental outcomes at two years of age.

### Material and methods

Prospective observational study performed in two hospitals between 2009 and 2011. Forty-three infants diagnosed with HIE within 6 hours of life were included. HIE was severe in 20 infants, moderate in 12, and mild in 11. Infants with moderate-to-severe HIE received whole-body cooling. Both the HIE cohort and a control group of 59 infants with suspected infection underwent measurement of CSF-NSE concentrations at between 12 and 72 hours after birth. aEEG monitoring was started at admission and brain MRI was performed within the first 2 weeks. Neurodevelopment was assessed at 24 months.

### Results

The HIE group showed higher levels of CSF-NSE than the control group: median 70 ng/ml (29; 205) *vs* 10.6 ng/ml (7.7; 12.9); p <0.001. Median levels of CSF-NSE in infants with severe, moderate, and mild HIE were 220.5 ng/ml (120.5; 368.8), 45.5 ng/ml (26, 75.3), and 26 ng/ml (18, 33), respectively. CSF-NSE levels correlated were significantly higher in infants with seizures, abnormal aEEG, or abnormal MRI, compared to those without abnormalities. Infants with an adverse outcome showed higher CSF-NSE levels than those with

**Funding:** The authors received no specific funding for this work.

**Competing interests:** The authors have declared that no competing interests exist.

normal findings (p<0.001), and the most accurate CSF-NSE cutoff level for predicting adverse outcome in the whole cohort was 108 ng/ml and 50ng/ml in surviving infants.

## Conclusions

In the era of hypothermia, CSF-NSE concentrations provides valuable information as a clinical surrogate of the severity of hypoxic-ischemic brain damage, and this information may be predictive of abnormal outcome at two years of age.

## Introduction

Despite the advent of therapeutic hypothermia (TH) as the sole specific intervention shown to improve neurodevelopmental outcomes, hypoxic-ischemic encephalopathy (HIE) continues to contribute to major worldwide perinatal mortality and to long-term disability in full-term and near-term newborns [1, 2].

Early and accurate assessment of the severity of brain damage after a perinatal hypoxic–ischemic event remains one of the most difficult challenges in neonatal care. A variety of clinical, neuroimaging, and neurophysiological tools, and combinations of such, are used to predict long-term outcome [3–7]. However, current methods of assessing the risk of brain injury in the newborn have inherent limitations during the first hours of life, and uncertainty regarding the severity of ongoing brain damage and eventual neurological outcome persists during this early period [8, 9].

A biochemical index of brain injury would be highly desirable to increase the reliability of predictions of neurological sequelae after hypoxic-ischaemic brain injury. Several central nervous system-specific molecules have been investigated in blood serum and in cerebrospinal fluid (CSF) as possible quantitative indexes of perinatal brain injury [10–14].

Neuron specific enolase (NSE) is a well-known biomarker of neuronal injury, both in adults and in pediatric patients [14–17]. In newborns with HIE, the concentrations of NSE have been measured in blood serum and in CSF [18–29]. Serum NSE levels in the setting of HIE after asphyxia show vast variability, which entails significant limitations for its use as a viable biomarker [25]. This variability is probably related to the inconsistent and unpredictable blood-brain-barrier permeability, the release of NSE from other extraneural sources in case of multiple organ injury, and confounding technical factors, such as the processing of the sample, the effect of hemolysis, the impact of the measuring method, and the of the reduction of concentrations as a function of storage time [30].

CSF levels of neuron-specific enolase (CSF-NSE) in infants with HIE have been shown to play a more consistent role as a surrogate marker of the extent of brain damage, and they provided good prediction of the outcome in infants with HIE in the pre-hypothermia era [20, 21, 26–28]. TH appears to lower CSF-NSE levels in infants with HIE; however, the predictive value of CSF-NSE for neurodevelopmental impairment at 12 months of age is not affected by cooling [29]. To our knowledge, no study has examined the correlation between CSF-NSE and other markers of ongoing brain damage during the acute phase of HIE, such us abnormal amplitude-integrated electroencephalogram (aEEG) and abnormal findings on cerebral MRI, nor its association with neurodevelopmental outcomes at 24 months in the hypothermia era.

The aim of this prospective cohort study of infants with HIE in the era of therapeutic hypothermia was to determine whether CSF-NSE correlates with the severity of HIE and other neonatal markers of ongoing brain damage, including aEEG and MRI, as well as with neurodevelopment at 2 years of age. This study shows that CSF-NSE within the first 72 hours may be a

useful biomarker for estimating ongoing brain damage and that it provides a surrogate outcome measure.

## Material and methods

The study population included infants with HIE consecutively born at ≥34 weeks gestational age with a birthweight ≥1800 grams, admitted to two universitary third-level hospitals in Barcelona, Spain, between April 2009 and July 2011. Infants were considered to have HIE if they met the following three criteria: 1) at least one of the following clinical surrogates of hypoxic-ischemic insult: altered fetal heart rate pattern, sentinel event, or labor dystocia; 2) Apgar score ≤5 at 5 and 10 minutes, need for resuscitation, including mask ventilation for more than 10 minutes after birth or endotracheal intubation, or acidosis (pH ≤7.0 and/or base deficit ≥16 mmol/L) in umbilical cord blood or within 60 minutes from birth; and 3) early neonatal encephalopathy defined as a syndrome of neurological dysfunction in the first 6 hours manifested by a subnormal level of consciousness with or without seizures (moderate or severe HIE) or palmary hyperexcitability, tremor, overactive myotatic reflexes, hypersensitivity to stimulation, or startle responses (mild HIE).

Newborns were excluded if a) they presented with congenital abnormalities, b) showed other identifiable etiologies of neurological dysfunction such as infection or genetic disease, or c) parental consent was not granted.

The severity of HIE was assessed immediately after admission, and always before starting TH, by three investigators (AGA, AA, GA); AGA was on call for the evaluation of infants of this project. Encephalopathy was classified as mild, moderate, or severe according to our semi-quantitative score which includes the aEEG findings [31, 32]. Newborns with moderate or severe HIE received whole-body cooling (Techotherm TSmed 200 N or Criticool, MTRE Ltd.) with rectal temperature maintained at 33–34˚C for 72 hours, following which they were slowly rewarmed (≤0.5˚C per hour). All cooled infants received sedation with fentanyl infusion throught the treatment.

### Neuro-specific enolase in cerebrospinal fluid

A CSF sample was obtained by lumbar puncture at between 12 and 72 hours of life. The greater value for CSF-NSE was chosen if two lumbar punctures were performed. The CSF-NSE concentrations were measured blind to clinical data, including the grading of HIE as well as the aEEG traces and the neuroimaging findings.

CSF aliquots of 0.4 ml were distributed in plastic tubes, which were immediately frozen and stored at -80ºC until analysed within 6 months after sampling. NSE concentrations in CSF were determined with an automated chemiluminescence immunoassay NSE Kritpor (Thermo Scientific® NSE Kryptor, BRAHMS GmbH., Germany). The coefficients of variation for interassay and intraassay variability were 2.8 to 6.2% and 2.2 to 9.7%, respectively. The sensitivity of this method was 1 ng/ml.

To compare CSF-NSE levels between patients with HIE and a control group, we included 59 patients without any abnormal neurological signs or confirmed central nervous system infections in whom lumbar puncture was performed as part of an early onset infection workup within 80 hours of life and for whom sufficient CSF was available for NSE analysis as part of a project on biomarkers in CSF.

### Amplitude-integrated electroencephalography (aEEG)

aEEG recordings were immediately started at admission in the NICU and continued for at least 24 hours in infants with mild HIE and throughout TH and rewarming in infants with

moderate to severe HIE. The aEEG recordings were collected using CFM 6000 (Olympic Medical, Natus Inc, Seattle, WA, USA) or Nicolet One TM System (Viasys Healthcare, San Diego, California, USA) and digitally stored. The traces were blindly assessed by one researcher (GA) attending to the background pattern, cyclicity, and seizure activity. Classification of aEEG background patterns included the following categories: continuous (CNV), discontinous (DCN), low voltage (LV), burst-supression (BS), and inactive or flat (FT) [33]. aEEG background trace was considered severely abnormal if BS, LV, or FT traces were observed. Electrical seizures were also noted.

## Neuroimaging studies

All MRI studies were performed in one of the two participating centers (HSJD). Brain magnetic resonance imaging (MRI) was performed using a 1.5 Tesla unit (General Electric) with a specific neonatal head coil. The imaging protocol has been published elsewhere [34]. Minimally, axial and sagittal T1-weighted, axial and coronal T2-weighted, and axial diffusion-weighted images were available for all patients. Some patients underwent an "early" study performed around the fourth day of life, and a "late" study during the second week of life. For infants with only one study, this was considered, while for infants with both an early and a late study, only the score of the late study was considered in establishing the correlation with CSF-NSE values.

Two researchers (TA, AGA) masked to clinical data and CSF-NSE levels reviewed MRIs. Images were scored according to the scheme previously reported by Rutherford et al [35, 36]. This score grades the severity of damage in four brain regions: the posterior limb of internal capsule (PLIC), the basal ganglia and thalami, the white matter, and the cortex. According to the findings, each site except for the PLIC is graded as normal, or as mildly, moderately, or severely affected. The PLIC is graded as normal, equivocal, or absent. Discrepancies in the scoring of the images were discussed and resolved by consensus. Moderate-to-severe injury was defined as moderate-to-severe damage in any of the regions examined. Infants were grouped into 4 patterns of injury: normal, basal ganglia/thalami injury (including basal ganglia/thalami ± cortical injury without white matter injury), watershed pattern (which included moderate to severe white matter injury without any other damage), and global injury (including minimally basal ganglia/thalami and white matter injury).

## Neurodevelopmental outcome

Neurodevelopmental assessments were made in all surviving infants at 24 months of age (median 24.3 months, IQR 23.3, 26.7 months, min-max 13–35 months) using the Bayley Scales of Infant and Toddler Development, Third Edition (BSID-III), which contains 3 individual developmental scores: a cognitive composite score, a language composite score (with receptive and expressive scores), and a motor composite score (with gross and fine motor sub-scores). These 3 scales have a mean score of 100 ± 15. A developmental quotient score < 70 (>2 SD below the mean) indicates significant delay, and a score of < 85 (>1 SD below the mean) indicates at least mild-to-moderate delay [37].

Cerebral palsy was defined according to the Surveillance of CP in Europe [38]. Adverse outcome was defined as death, cerebral palsy, or Bayley-III cognitive, language, or motor scores <85.

## Statistical analysis

Regarding perinatal, neurological, and outcome findings, qualitative variables were summarized with absolute and relative frequencies, and qualitative variables with median and

interquartile range (IQR) or mean (standard deviation). Continuous variables were compared using Mann-Whitney's *U* test or Kruskal-Wallis test, as appropriate. Categorical variables were compared using the Chi-squared test or Fisher's exact test.

CSF-NSE values were log-transformed to approximate a normal distribution and to achieve homogeneity of variance. Then differences between groups were tested by analysis of variance, with Tukey *post hoc* pairwise analysis. Spearman correlation coefficients ($r_s$) were used to assess the correlation between quantitative variables. Regression models were used to evaluate the relationship between biomarker levels and the presence and severity of HIE (controls, and mild, moderate, and severe HIE), including interaction for the age at which lumbar puncture was performed. In order to estimate the predictive capacity of the CSF-NSE regardless of the degree of HIE and the age at lumbar puncture, the CSF-NSE-outcome association was estimated by multiple regression, adjusting for these covariates. To determine the optimal cutoff value for CSF-NSE, defined as the highest value of sensitivity multiplied by specificity, receiver operating characteristic curve (ROC) analysis for CSF-NSE levels was made. The area under the curve, sensitivity (S), specificity (Sp), positive predictive value (PPV), and negative predictive value (NPV), including 95% confidence intervals (95% CIs) were expressed. All reported P values were 2-sided, and a P value of <0.05 was considered as indicating statistical significance. Statistical analyses were performed using SPSS verson 20 (IBM, Armonk, NY, USA).

We planned to recruit 50 subjects over a 2-year period. Accounting for 10–15% patient loss for analysis due to clinical instability of the patient, haemorrhage, or insufficient CSF, *a priori* power calculations were based on analysis of data from 40 subjects. This sample size provided at least 90% statistical power (two-sided alpha risk 5%) to detect a minimum expected effect size of 50 points (SD 48) of difference in biomarker levels between patients with adverse neurological outcome and those with non-adverse neurological outcome.

## Ethical considerations

Written information was given to the parents and written consent was obtained at the bedside on admission of each infant after explanation of the study and before its onset. The research was conducted according to the Declaration of Helsinki principles and those of the human studies committee that approved the study (Clinic Hospital Protocol Number: HBC/2010/6049).

## Results

Fifty-nine consecutive patients with HIE were recruited during the study period but 6 were excluded due to perinatal infection with *Listeria monocytogenes* (2 infants), multiple hemorrhagic strokes (1 infant), spinal cord injury (1 infant), late admission at 5 days of age (1 infant), or lack of parental informed consent (1 infant). Of the 53 candidate patients, CSF-NSE could not be performed in 10 infants due to clinical instability, haemorrhage, or insufficient CSF.

CSF-NSE was performed at a median age of 48 hours (15, 73) in the group of infants with HIE and 32 hours (24, 58) in the control group; p = 0.344. Compared to controls, patients with HIE had fewer spontaneous deliveries (7% vs 61%), lower cord pH values (6.94 ± 0.16 vs 7.22 ± 0.11), and lower Apgar scores at 5 minutes (5 (3, 6) vs 10 (8, 10)). Birthweight was lower in the HIE group compared to controls: 3004 ± 587 grams vs 3266 ± 489 grams (p = 0.034). No differences were found between the HIE and control groups regarding gestational age or gender.

The main perinatal characteristics of the 43 recruited infants with HIE, according to the severity of encephalopathy, are shown in Table 1. All infants with moderate or severe HIE were cooled, except for 3 infants with severe HIE, due to late transfer beyond 12 hours of life

**Table 1. Perinatal, neurological and outcome findings of the 43 HIE infants enrolled in the study.**

|  | Total HIE N = 43 | Mild HIE N = 11 | Moderate HIE N = 12 | Severe HIE N = 20 | P value |
|---|---|---|---|---|---|
| Gestational age, weeks | 38 (37, 40) | 38 (37, 40) | 39 (37, 41) | 38 (36, 40) | 0.685 |
| Weight, grams | 3000 (2530, 3500) | 3160 (2600, 3700) | 3115 (2423, 3498) | 2835 (2500, 3355) | 0.346 |
| Sex (female) | 18/43 (42) | 4/11 (36) | 5/12 (42) | 9/20 (45) | 0.897 |
| Intrauterine growth retardation | 4/43 (9) | 0/11 | 2/12 (17) | 2/20 (10) | - |
| Sentinel event | 16/43 (37) | 4/11 (36) | 4/12 (33) | 8/20 (40) | 0.929 |
| Spontaneous delivery | 3/43 (7) | 1/11 (9) | 1/12 (8) | 1/20 (5) | - |
| Umbilical cord pH[a] | 6.97 (6.81, 7.03) | 7.00 (6.99, 7.05) | 6.90 (6.79, 7.00) | 6.94 (6.81, 7.06) | 0.190 |
| Apgar score at 5 minutes | 5 (3, 6) | 6 (4, 7) | 5.5 (4, 7) | 4 (2, 5) | 0.051 |
| Advanced resuscitation[b] | 31/43 (72) | 6/11 (55) | 8/12 (67) | 17/20 (85) | - |
| Electrical seizures during TH | 24/43 (56) | 0/11 (0) | 6/12 (50) | 18/20 (90) | <0.001 |
| Abnormal aEEG BGP in first 6 hours of life | 20/39 (51) | 0/8 (0) | 6/12 (50) | 14/19 (74) | <0.001 |
| Abnormal aEEG BGP in first 80 hours of life[c] | 24/43 (56) | 0/11 (0) | 6/12 (50) | 18/20 (90) | <0.001 |
| MRI Score | 2.5 (0, 9.3) | 0 (0, 1) | 0 (0, 4) | 9.5 (6.5, 11) | <0.001 |
| Moderate-severe injury (MRI) | 20/38 (51) | 1/11 (9) | 3/11 (27) | 16/16 (100) | <0.001 |
| Global injury pattern (MRI) | 17/38 (45) | 0/11 (0) | 2/11 (18) | 15/16 (94) | <0.001 |
| Death | 14/43 (33) | 0/11 | 1/12 (8) | 13/20 (65) | <0.001 |
| Adverse outcome[c] | 22/42 (52) | 0/11 | 3/11 (27) | 19/20 (95) | <0.001 |
| Cerebral palsy | 5/28 (18) | 0/11 | 1/10 (10) | 4/7 (57) | 0.006 |
| CP and/or any abnormal BSID-III score < 85[d] | 8/28 (29) | 0/11 | 2/10 (20) | 6/7 (86) | <0.001 |
| CP and/or any abnormal BSID-III score < 70[d] | 5/28 (18) | 0/11 | 0/10 | 3/7 (43) | 0.006 |
| BSID-III Score (Motor)[e] | 94 (83, 99) | 99 (94, 103) | 96 (90, 98) | 79 (45, 88) | 0.009 |
| BSID-III Score (Cognitive)[e] | 98 (86, 105) | 105 (95, 105) | 105 (90, 110) | 80 (45, 85) | 0.006 |
| BSID-III Score (Language)[e] | 91 (83, 100) | 97 (91, 100) | 91 (88, 103) | 77 (45, 83) | 0.019 |

Quantitative variables are expressed as median (interquartile range), and categorical values are expressed as n/N (%). Continuous variables were compared using Kruskal-Wallis test. Categorical variables were compared using the Chi-squared test. P value of <0.05 was considered as indicating statistical significance. aEEG: amplitude integrated electroencephalogram; BGP: background pattern; BSID-III: Bayley Scales of Infant and Toddler Development Third Edition; CP: cerebral palsy; MRI: magnetic resonance imaging; TH: therapeutic hypothermia.

[a]Data were not available in 1, 2, and 3 infants with mild, moderate, and severe HIE, respectively.

[b]Advanced resuscitation: tracheal intubation, chest compressions and/or adrenaline.

[c]Adverse outcome: death or cerebral palsy and/or BSID-III score <85.

[d]Outcome in 28/29 surviving infants.

[e]BSID-III test was performed in 24 out of 29 surviving infants

in 2 infants and to critically ill status in one case. Infants with mild HIE were not cooled. MRI was performed in 37 patients, as five patients died before an MR scan could be performed (4 severe and one moderate HIE). Both aEEG background pattern and MRI findings were associated with severity of HIE (Table 1). Brain MRI was performed at a median age of 10 days; IQR 4.4, 13.9.

Fourteen infants (33%) did not survive the neonatal period, one with moderate HIE and 13 with severe HIE. In the only patient with moderate HIE who died, death was due to severe coagulopathy. In the 13 infants with severe HIE, death was preceded by an end-of-life decision (EoL). The EoL decision was considered for those patients with persistent severe encephalopathy (coma) in combination with persistent severe altered aEEG and severe neuroimaging findings (brain ultrasound scans and/or MRI). CSF-NSE levels were not considered in making an end-of-life decision.

Of the 29 neonates who survived, 24 were evaluated with BSID-III; additionally, 4 were contacted by phone, with parents reporting that they had a normal outcome. One infant were lost

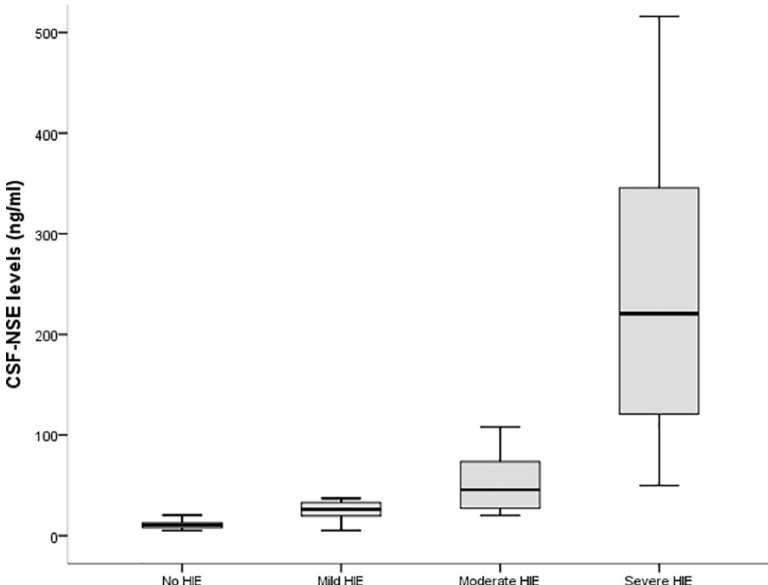

**Fig 1. Cerebrospinal fluid levels of neuron-specific enolase in 59 controls and 43 infants with HIE, according to the severity of the encephalopathy.** CSF-NSE levels in the control group were 10.6 (7.7; 12.9). CSF-NSE levels in 20, 12, and 11 infants with severe, moderate, and mild HIE, respectively, were 220.5 ng/ml (120.5; 368.8), 45.5 ng/ml (26, 75.3), and 26 ng/ml (18, 33). No statistically significant differences were found between any of the grades of HIE compared to controls or between any of the grades of HIE (analysis of variance after log-transformation of CSF-NSE values, with Tukey *post hoc* pairwise analysis); p<0.05. HIE: hypoxic-ischaemic encephalopathy; CSF-NSE: cerebrospinal fluid–neuron specific enolase.

to follow-up. Of the overall cohort, 22 infants (54%) had adverse outcomes, and 8 infants out of the 29 neonates who survived had CP and/or BSID-III Score < 85. Table 1 shows the outcome results according to the severity of HIE.

## CSF-NSE and neurological variables

The HIE group showed significantly higher levels of CSF-NSE than the control group: median 70 ng/ml (29; 205) *vs* 10.6 (7.7; 12.9); p <0.001. According to the severity of HIE, all subgroups presented higher CSF-NSE levels than the control population (p<0.001). Infants with severe HIE had higher levels of CSF-NSE than those with moderate HIE (p<0.001), and infants with moderate HIE had higher levels than those with mild HIE (p = 0.024). CSF-NSE levels in infants with severe, moderate, and mild HIE were: 220.5 ng/ml (120.5; 368.8), 45.5 ng/ml (26, 75.3) and 26 ng/ml (18, 33), respectively (Fig 1). All severe cases had CSF-NSE levels above 109 ng/ml except for two infants with levels of 50 ng/ml and 70 ng/ml, respectively. All mild cases had a level below 38 ng/ml.

Regression analysis showed that age at lumbar puncture did not influence the association between CSF-NSE levels and the presence or severity of HIE.

CSF-NSE concentrations were higher in infants with abnormal neurological findings including seizures, aEEG background activity, and MRI, compared to those without abnormalities (p<0.001) (Table 2).

Regarding MRI injury, the 20/38 infants that had moderate-to-severe injury had higher CSF-NSE levels compared to the infants who did not have injury (15/38) or those with mild injury (3/38): 133 (73; 268) vs 28.5 (20.8; 36.3) (p<0.001). The pattern of damage on MRI was predominantly global (17/38), and only 2 and 3 infants out of the 38 infants with MRI had solely basal ganglia/thalamus and watershed injury, respectively. Infants with global injury had

**Table 2. Cerebrospinal fluid levels of neuron-specific enolase of 43 infants with HIE according to their neurological findings and outcomes.**

| Variables | Yes | | No | | P value |
|---|---|---|---|---|---|
| | n | CSF-NSE (ng/ml) | n | CSF-NSE (ng/ml) | |
| Moderate or severe encephalopathy | 32 | 116.5 (56, 267.8) | 11 | 26 (18, 33) | <0.001 |
| Electrical seizures during TH | 24 | 133 (71.8, 267.8) | 19 | 28 (21, 37) | <0.001 |
| Abnormal aEEG BGP in first 6 hours of life | 20 | 133 (77.8, 368.8) | 19 | 33 (25, 59) | <0.001 |
| Abnormal aEEG BGP in first 80 hours of life | 24 | 183.5 (87,5; 294.5) | 19 | 32 (23; 50) | <0.001 |
| Moderate-severe injury (MRI) | 20 | 133 (72.5, 267.8) | 18 | 28.5 (20.8, 36.3) | <0.001 |
| Global injury pattern (MRI) | 17 | 166 (96.5, 274.5) | 21 | 29 (22, 46) | <0.001 |
| Cerebral palsy | 5 | 268 (130; 290) | 23 | 33 (23; 59) | 0.002 |
| CP and/or BSID-III Score < 85[a] | 8 | 157.5 (56, 277.8) | 20 | 28.5 (21.5, 37) | 0.001 |
| CP and/or BSID-III Score < 70[a] | 5 | 268 (130, 290) | 23 | 33 (23, 59) | 0.002 |
| Adverse outcome[b] | 22 | 203 (112.3; 322.3) | 20 | 29 (21.5; 36.8) | <0.001 |
| Death | 14 | 218.5 (121.5, 483) | 29 | 34 (25, 73.5) | <0.001 |

CSF-NSE values are expressed as median (interquartile range). Mann-Whitney U test was used to analyze the differences in CSF-NSE levels between groups. P value of <0.05 was considered as indicating statistical significance. aEEG: amplitude integrated electroencephalogram; BGP: background pattern; BSID-III: Bayley Scales of Infant and Toddler Development Third Edition; CP: cerebral palsy; MRI: magnetic resonance imaging; TH: therapeutic hypothermia.

[a]Outcome in 28/29 surviving infants

[b]Adverse outcome: death or cerebral palsy and/or BSID-III score <85.

higher CSF-NSE levels: 166 (96.5, 275) vs 29 (22, 46) (p<0.001). CSF-NSE levels showed a positive correlation with the MRI score: $r_s$ 0.824, p<0.001.

## CSF-NSE and outcome

Infants with an adverse outcome showed higher CSF-NSE levels than those with normal findings (p< 0.001) (Fig 2 and Table 2). ROC analysis revealed a CSF-NSE cutoff value of maximum predictive accuracy of 108 mg/dL (AUC 0.97, S 0.86, Sp 1.0, PPV 1.0, NPV 0.87) for prediction of adverse outcome. Among surviving infants, CSF-NSE optimal cutoff-point to predict CP and/or BSID-III score <85 was 50 mg/dL (AUC 0.93, S 1.0, Sp 0.8, PPV 0.67, NPV 1.0) (Table 3).

CSF-NSE levels showed a moderate negative correlation with the three BSDI-III domains (motor, cognitive, and language composite scores): $r_s$ -0.650, p = 0.001; $r_s$ -0.641, p = 0.001; and $r_s$ -0.604, p = 0.002 respectively.

Regression analysis showed that age at lumbar puncture nor the severity of HIE did not interact with the association between CSF-NSE level and outcome variables.

The predictive values for the neurological variables included in the study (severity of HIE, aEEG background pattern, MRI findings, and the CSF-NSE cutoff-point of 108 ng/ml) are shown in Table 4.

## Discussion

The study shows that CSF-NSE within the first 72 hours may be a useful biomarker for estimating ongoing brain damage, based on its correlation with the clinical grading of encephalopathy, aEEG tracings, and MRI findings. More importantly, NSE concentrations in CSF can provide a surrogate outcome measure.

Several methods have been used for early estimation of the cerebral damage after hypoxic-ischemic injury and prediction of outcome. The most readily available tools include clinical grading of the encephalopathy, electrophysiological information such us assessment of

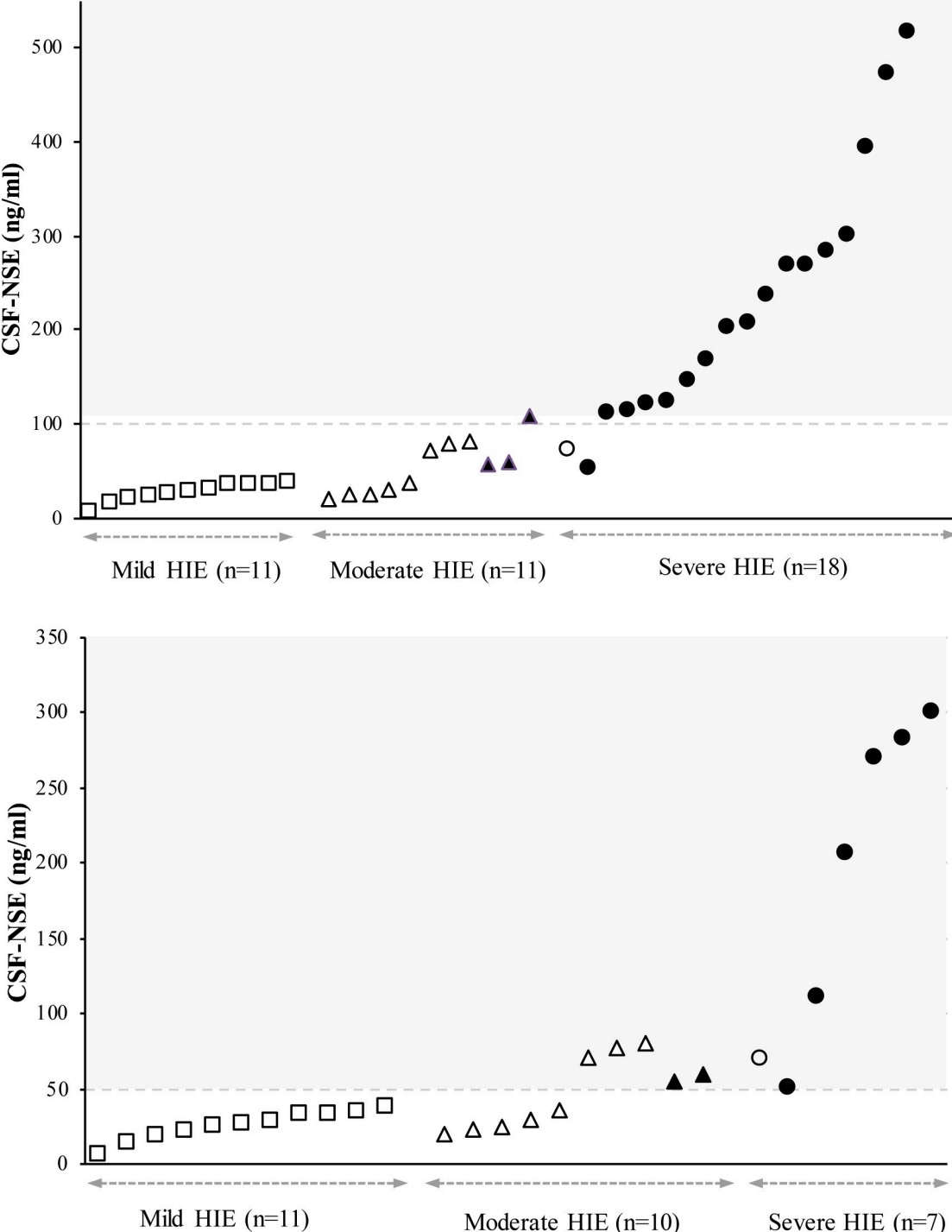

**Fig 2. Cerebrospinal fluid levels of neuron-specific enolase of 43 patients according to their neurological findings and outcomes.** Each CSF-NSE value represents a square-shaped infant, trigulum, or circle according to the severity of HIE (mild, moderate, or severe). Filled shapes represent infants with adverse ouctomes (death or cerebral palsy and/or BSID-III score < 85). Empty dots represent those infants without adverse outcome. The figure above shows the entire cohort of 43 children, and the one below shows the 29 surviving infants. Two infants with very severe HIE with extreme CSF-NSE values (2855 ng/ml and 1490 ng/ml) and one infant lost in the follow-up are not represented in the figures. The dashed line represents the cutoff value of 108 mg/dL and 50 mg/dl for adverse outcome, according to the maximum predictive accuracy in the receiver operating curve analysis, for the total cohort (figure above) and only the surviving infants (figure below), respectively. HIE: hypoxic-ischaemic encephalopathy; CSF-NSE: cerebrospinal fluid–neuron specific enolase; aEEG: amplitude integrated electroencephalogram; BGP: background

pattern; BSID-III: Bayley Scales of Infant and Toddler Development Third Edition; CP: cerebral palsy; MRI: magnetic resonance imaging.

electrical activity by means of an amplitude-integrated EEG and evoked responses, neuroimaging, and cerebral blood flow studies [7, 9, 39–41].

However, all these evaluations have some limitations during the first 72 hours. During this time frame, the level of alertness is often confounded by the use of sedatives during cooling and/or antiepileptic drugs if clinical or electrical seizures occur, rendering clinical evaluation less reliable [42, 43]. In addition, qualitative evaluation of significant brain injury on both diffusion-weighted and conventional MRI during the first 48h may be underestimated [2, 43, 44]. Other quantitative techniques such as quantitative diffusion parameters and proton magnetic resonance spectroscopy have been used to overcome these limitations [45]. However, in clinical practice their use in infants with perinatal HIE during this narrow ~~short~~ time frame is limited. Sick infants may not tolerate transfer to radiology rooms, and early MRI is challenging as it requires specialized interpretive expertise.

Amplitude-integrated EEG background patterns in neonatal encephalopathy correlate closely with neurological outcome [46]. However, their predictive value depends on how long the background patterns are altered in relation to the hypoxic-ischemic insult, and the greater predictive value for adverse outcome peaks at 48–72 hours of life in cooled infants [41].

**Table 3. Receiver operating characteristic curve analysis of cerebrospinal fluid levels of neuron-specific enolase and neurological findings.**

| Variable | N | Cutoff-point | Area under the ROC (95% CI) | Sensitivity (95% CI) | Specificity (95% CI) | PPV (95%CI) | NPV (95%CI) |
|---|---|---|---|---|---|---|---|
| Moderate-to-severe HIE | 32 | 50 | 0.91 (0.83,0.99) | 0.81 (0.64,0.93) | 1.00 (0.72,-) | 1.00 (0.86,1.00) | 0.65 (0.42,-) |
| Severe HIE | 20 | 110 | 0.98 (0.95,1.00) | 0.90 (0.68,0.98) | 1.00 (0.85,-) | 1.0 (0.82,1.00) | 0.92 (0.73,-) |
| Electrical seizures during TH | 24 | 50 | 0.85 (0.71,0.99) | 0.92 (0.73,0.99) | 0.79 (0.54,0.94) | 0.85 (0.64,0.98) | 0.88 (0.65,0.97) |
| Abnormal aEEG BGP in first 6 hours of life | 20 | 77 | 0.87 (0.75,0.99) | 0.80 (0.56,0.94) | 0.89 (0.67,0.99) | 0.89 (0.65,0.97) | 0.80 (0.58,0.97) |
| Abnormal aEEG BGP in first 80 hours of life | 24 | 77 | 0.92 (0.83,1) | 0.83 (0.63,0.95) | 0.95 (0.74,0.99) | 0.95 (0.76,0.99) | 0.82 (0.60,0.99) |
| Moderate-severe injury (MRI) | 20 | 80 | 0.93 (0.85,1.00) | 0.75 (0.51,0.91) | 1.00 (0.81,-) | 1.00 (0.79,1.00) | 0.78 (0.55,-) |
| Global injury pattern (MRI) | 17 | 80 | 0.95 (0.88,1.00) | 0.82 (0.57,0.96) | 0.95 (0.76,0.99) | 0.93 (0.69,0.99) | 0.87 (0.65,0.99) |
| Cerebral palsy | 5 | 205 | 0.95 (0.84,1.00) | 0.80 (0.28,0.99) | 1.00 (0.85,-) | 1.00 (0.50,1.00) | 0.96 (0.69,-) |
| CP and/or BSID-III Score < 85[a] | 8 | 50 | 0.93 (0.83,1.00) | 1.00 (0.63,-) | 0.80 (0.56,0.94) | 0.67 (0.39,-) | 1.00 (0.77,1.00) |
| CP and/or BSID-III Score < 70[a] | 5 | 205 | 0.95 (0.84,1.00) | 0.80 (0.28,0.99) | 1.00 (0.85,-) | 1.00 (0.49,1.00) | 0.96 (0.69,-) |
| Adverse outcome[b] | 22 | 108 | 0.97 (0.93,1.00) | 0.86 (0.65,0.97) | 1.0 (0.83,-) | 1.0 (0.82,1.0) | 0.86 (0.66,-) |
| Death | 14 | 108 | 0.91 (0.83,0.99) | 1.00 (0.77,-) | 0.83 (0.64,0.94) | 0.74 (0.51,-) | 1.00 (0.85,1.00) |

aEEG: amplitude integrated electroencephalogram; BGP: background pattern; BSID-III: Bayley Scales of Infant and Toddler Development Third Edition; CI: confidence interval; CP: cerebral palsy; MRI: magnetic resonance imaging; NPV: negative predictive value; PPV: positive predictive value; ROC: Receiver operating curve; TH: therapeutic hypothermia.

[a]Outcome in 28/29 surviving infants.

[b]Adverse outcome: death or cerebral palsy and/or BSID-III score <85. CSF-NSE values are expressed in ng/ml

**Table 4. Predictive values of the different neurological tools related to the outcome.**

| Adverse outcome | Sensitivity | Specificity | PPV | NPV | Odds Ratio (95%CI) |
|---|---|---|---|---|---|
| Severe HIE | 0.86 | 0.95 | 0.95 | 0.86 | 120.3 (11.5;1262.8) |
| Moderate-to-severe | 1.00 | 0.55 | 0.71 | 1.00 | 54.5 (2.9;1021.5) |
| Abnormal aEEG BGP in first 80 hours of life | 0.86 | 0.75 | 0.79 | 0.83 | 19 (3.9;92.6) |
| Moderate-severe injury (MRI) | 0.94 | 0.85 | 0.84 | 0.94 | 90.7 (8.5;964.0) |
| Global injury pattern (MRI) | 0.82 | 0.9 | 0.88 | 0.86 | 42 (6.2,286.7) |
| CSF-NSE > 108 ng/ml | 0.82 | 1.00 | 1.00 | 0.83 | 168.6 (8.5,3347.3) |

aEEG: amplitude integrated electroencephalogram; BGP: background pattern; CI: confidence interval; MRI: magnetic resonance imaging; NPV: negative predictive value; PPV: positive predictive value; ROC: Receiver operating curve; TH: therapeutic hypothermia.
[a]Adverse outcome: death or cerebral palsy and/or BSID-III score <85.

Despite the limitations in accurate assessment of brain damage during the first 72 hours after the hypoxic-ischemic insult, this challenge has to be addressed in order to provide early neuroevolutive prognostic information to parents [9].

The concept of "window of opportunity" is used in severe brain injury to indicate as the period of time in which a critically ill neonate might die after removal from the ventilator. After this window, the infant would no longer be dependent on such treatment and would probably survive with subjectively intolerable disability. This window of opportunity is a determining factor in the urgency of having neurological outcome prognostic indicators during the first 72 hours [47].

To improve early identification of infants at risk of neuroevolutive sequelae, reliable early biomarkers of brain damage are a valuable clinical tool to improve accurate identification of infants at risk after a hypoxic-ischemic insult.

Neuron specific enolase (NSE) is a well-known biochemical marker of neuronal injury in adults [16, 17], and pediatric patients [14, 15]. NSE is stable in biological fluids, it originates predominantly in the cytoplasm of neuroendocrine cells and neurons, and its measurement does not elicit immunologic cross-reactivity with non-neuronal enolase.

For clinical purposes, measurement of this potentially sensitive biomarker for brain damage in peripheral blood may have advantages compared to CSF. Blood sampling is less invasive, and can be performed more often, even when the patient is unstable. However, serum concentration both in the pre-hypothermia era and in cooled infants has shown inconsistent results in relation to the severity of brain injury assessed by clinical grading of HIE, neuroimaging, and outcome beyond 12 months of age [18–25]. There are several possible explanations for these controversial results. First, brain injury does not necessarily result in consistent, predictable blood-brain-barrier disruption. Second, increased serum NSE is not necessarily of CNS origin. Several tissues, including muscle, kidney, heart, adipose tissue, adrenal glands, lymphocytes, red blood cells, and platelets contain significant amounts of NSE [48–52]. A hypoxic-ischemic event can result in multiple organ injury, including all those tissues and disseminated intravascular coagulation. Finally, technical confounders must be taken into account. These include the processing of the sample, the effects of the time on the sampling stored since after 6 months at -80ºC there might be a significant decrease in concentrations in CSF, and, particularly, the effect of hemolysis [53]. Moreover, the choice of measuring methods may have influence on the levels of NSE [54], so care should be taken in using our specified cut-off level of NSE for determination of prognosis if another NSE determination method was used.

The main advantages of CSF sampling is the fact that the brain is in direct contact with the CSF and the damaged brain tissue releases high quantities of proteins into this fluid. In

addition, there are no sources of NSE other than neurons [30]. Hence, the consistent value of CSF-NSE levels in the first 96 hours of life seen in the prehypothemia era are not surprising, and this biomarker reflects brain damage in accordance with the clinical severity of HIE, as do MRI and aEEG findings [14, 20, 26, 28].

To our knowledge only one study has been published on CSF-NSE that included cooled infants with HIE, but this study did not examine the relation between CSF-NSE concentrations and aEEG and MRI findings as surrogates for the severity of hypoxic-ischemic event [29]. We examined the relationship between CSF-NSE and relevant clinical variables (seizures), in addition to aEEG, MRI, and neurodevelopmental outcomes. Our results are consistent with the previously reported value of the CSF-NSE to identify severe brain damage after sudden and unexpected postnatal collapse [55], and with the relationship between CSF-NSE levels and infarction volume and topology, as well as neurodevelopment at 2 years of age in infants with neonatal arterial ischemic stroke [56].

One strength of our study is that it offers cutoff points that are relevant in clinical practice. CSF-NSE levels $\geq$ 108 ng/ml showed the greatest specificity and positive predictive value for identifying infants at increased risk of adverse outcome. Considering only those who survived, optimal cutoff point to predict abnormal outcome was 50 ng/ml. CSF-NSE levels may be particularly useful when additional information is needed about brain injury and prognosis, especially if there are inconsistencies between prognostic tools, such as neurological examination and neurophysiological studies, and also if an end-of-life decision is considered within the first 72 hours of life [8].

Biochemical markers of brain injury, such as NSE, are released in a time sequence related to brain damage. In animal models of focal and global ischemia, an early peak of serum NSE levels reflects the ischemia-induced cytoplasmic loss of NSE in neurons that is detectable before irreversible neuronal damage occurs [57, 58]. This has also been corroborated in human adults with stroke, where, after an initial rise within 3 hours, NSE concentrations in blood serum decrease followed by a secondary increase until day 5, with concentrations after 24 hours reflecting the volume of infarcted brain areas [59, 60].

Several studies have determined serum NSE at different times in the first days of life. These studies have shown that serum NSE concentrations are relatively stable and higher during the first 24 hours after birth, decreasing thereafter [24, 61]. It is not known how soon NSE can be detected in the CSF after brain injury, nor are the kinetics of its release and clearance in this biological fluid well understood. Therefore, the time point of maximum diagnostic efficacy is unknown [49]. The scheduling of CSF sampling is generally decided arbitrarily, but in most studies lumbar puncture has been performed within the first 72 hours of life [21, 26, 28, 62]. Despite a range of 12–80 hours in our study, we did not find that the age at lumbar puncture was an interaction factor between the CSF-NSE levels and the variables analyzed. Whether serial measurements might provide indirect information regarding ongoing neuronal injury and might therefore help increase accuracy in predicting patient outcome is unknown. However, lumbar puncture is an interventional procedure, and it is not free from risks. It should not be performed when the infant is unstable or suffers from significant coagulopathy, which makes it difficult to obtain serial samples.

Our determination of CSF-NSE concentrations at a median age of 48 hours after birth may mirror irreversible neuronal damage as it is correlated with surrogate biomarkers of brain damage and the outcomes in this study. The reliability of CSF-NSE to predict subsequent adverse neurological outcomes was similar to the reliability of aEEG and neuroimaging. Therefore, CSF-NSE testing is suggested as both a valuable diagnostic tool in clinical management of the infant with HIE and a prognostic parameter during the course of the encephalopathy. In addition to helping to predict neurodevelopmental outcomes, biochemical brain

damage biomarkers such as CSF-NSE should gain momentum in clinical research on HIE. We feel that they can be particularly useful to identify newborns at 48 hours who are at risk of adverse prognosis and potential candidates to receive adjuvant neuroprotective therapies to promote neuronal regeneration and structural recovery from injury during the tertiary phase of HIE (as is the case with stem cell transplantation and anti-inflammatory agents and ganglio-sides). Furthermore, this biochemical marker could be useful for rigorous monitoring of thera-peutic interventions aimed at ameliorating brain damage.

Our study has some limitations. Like many longitudinal studies on neurobiological markers that may reflect the severity and progression of brain injury, the inability to obtain or analyze some CSF samples was a concern. Given the low incidence and prevalence of HIE, one for every 1000 births in our setting [63], the number of recruited infants was not huge. Larger vali-dation studies would be advisable. Our study was not powered to evaluate death or disability separately, and since most infants with severe damage die, larger studies are needed to assess the relationship between biomarkers and outcome, independent of mortality. This is especially important for the group of infants with moderate HIE—the group with the greatest prognostic uncertainty. In our series 3 of the 32 infants with moderate-to-severe HIE were not cooled; all three had severe HIE. When analyses were performed eliminating these three patients, no dif-ference in results emerged, even when including optimal cut-off values for CSF-NSE (ROC analysis) (S1 and S2 Tables). Finally, Since it is not ethical to perform lumbar puncture in healthy babies, we decided to use babies with suspected infection as control subjects. This approach has already been used in previous studies with the same purpose as ours, and we believe that CSF-NSE values are highly unlikely to be influenced by the consideration of infec-tious risk since their neurological status and CSF findings were normal [21, 26, 27].

## Conclusion

HIE is a rare but potentially devastating disease associated with high morbidity and mortality. Our study suggests that NSE in CSF measured in infants with HIE at a median age of 48 hours of life might be a useful tool to identify those newborns with suspicion of severe brain injury based on clinical assessment, aEEG, and MRI. Furthermore, in the era of hypothermia, this biomarker appears to be able to distinguish infants will fully recover from those who are likely to present an adverse outcome.

## Supporting information

**S1 Table. Cerebrospinal fluid levels of neuron-specific enolase of the 40 cooled infants with HIE according to their neurological findings and outcomes.** The three infants that did not undergo hypothermia treatment had severe HIE and were not included in this analysis (NSE values were 166, 281, and 2855 ng/ml). CSF-NSE values are expressed as median (inter-quartile range). Mann-Whitney U test was used to analyze the differences in CSF-NSE levels between groups. P value of <0.05 was considered as indicating statistical significance. aEEG: amplitude integrated electroencephalogram; BGP: background pattern; BSID-III: Bayley Scales of Infant and Toddler Development Third Edition; CP: cerebral palsy; MRI: magnetic reso-nance imaging; TH: therapeutic hypothermia. [a]Outcome in 28/29 surviving infants. [b]Adverse outcome: death or cerebral palsy and/or BSID-III score <85.
(DOCX)

**S2 Table. Receiver operating characteristic curve analysis of cerebrospinal fluid levels of neuron-specific enolase and neurological findings of the 40 cooled infants.** The three infants that did not undergo hypothermia treatment had severe HIE and were not included in

this analysis (NSE values were 166, 281 and 2855 ng/ml). aEEG: amplitude integrated electro-encephalogram; BGP: background pattern; BSID-III: Bayley Scales of Infant and Toddler Development Third Edition; CP: cerebral palsy; MRI: magnetic resonance imaging; PPV: positive predictive value; NPV: negative predictive value; ROC: Receiver operating curve; TH: therapeutic hypothermia. [a]Outcome in 28/29 surviving infants. [b]Adverse outcome: death or cerebral palsy and/or BSID-III score <85. CSF-NSE values are expressed in ng/ml. (DOCX)

**S1 Dataset.**
(XLSX)

# Acknowledgments

We are grateful to Mrs. Sara Calvo, Dr. Isabel Benavente-Fernandez and Dr. Carlos Ochoa for their help with statistical analysis.

# Author Contributions

**Conceptualization:** Alfredo Garcia-Alix.

**Data curation:** Marisol-Zulema León-Lozano, Gemma Arca, Thais Agut, Ana Alarcón, Alfredo Garcia-Alix.

**Formal analysis:** Juan Arnaez.

**Investigation:** Marisol-Zulema León-Lozano, Juan Arnaez, Ana Valls, Gemma Arca, Thais Agut, Ana Alarcón, Alfredo Garcia-Alix.

**Methodology:** Ana Valls, Alfredo Garcia-Alix.

**Supervision:** Juan Arnaez, Alfredo Garcia-Alix.

**Validation:** Juan Arnaez.

**Visualization:** Marisol-Zulema León-Lozano, Juan Arnaez, Alfredo Garcia-Alix.

**Writing – original draft:** Marisol-Zulema León-Lozano.

**Writing – review & editing:** Juan Arnaez, Ana Valls, Gemma Arca, Thais Agut, Ana Alarcón, Alfredo Garcia-Alix.

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
