## [Decision Letter · Decision Letter 0]

3 Mar 2020

PONE-D-19-29027

Cerebrospinal fluid levels of neuron-specific enolase predict the severity of brain damage in newborns with neonatal hypoxic-ischemic encephalopathy treated with hypothermia

PLOS ONE

Dear Dr Garcia-Alix,

Thank you for submitting your manuscript to PLOS ONE. After careful consideration, we feel that it has merit but does not fully meet PLOS ONE’s publication criteria as it currently stands. Therefore, we invite you to submit a revised version of the manuscript that addresses the points raised during the review process.

Please address the comments raised by both Reviewers, especially the major comments in order to further improve your manuscript.

We would appreciate receiving your revised manuscript by Apr 17 2020 11:59PM. To enhance the reproducibility of your results, we recommend that if applicable you deposit your laboratory protocols in protocols.io, where a protocol can be assigned its own identifier (DOI) such that it can be cited independently in the future. For instructions see: http://journals.plos.org/plosone/s/submission-guidelines#loc-laboratory-protocols

We look forward to receiving your revised manuscript.

Kind regards,

Pierre Gressens

Academic Editor

PLOS ONE

Journal Requirements:

2) Please include captions for your Supporting Information files at the end of your manuscript, and update any in-text citations to match accordingly. Please see our Supporting Information guidelines for more information: http://journals.plos.org/plosone/s/supporting-information.

3) We noticed you have some minor occurrence of overlapping text with the following previous publication(s), which needs to be addressed:

https://www.pedneur.com/article/S0887-8994(18)31326-2/fulltext

The text that needs to be addressed is in the Discussion section.

In your revision ensure you cite all your sources (including your own works), and quote or rephrase any duplicated text outside the methods section. Further consideration is dependent on these concerns being addressed.

4) Please change your reference to "p=0.000" to "p<0.001" or as similarly appropriate, as p values cannot equal zero.

5) Thank you for your ethics statement : "Written information was given to the parents and written consent was obtained at the bedside on admission of each infant after explanation of the study and before its onset. The Human Studies Committee of each institution approved the study"

6) We note you have included a table to which you do not refer in the text of your manuscript. Please ensure that you refer to Table 3 in your text; if accepted, production will need this reference to link the reader to the Table.

7) Thank you for stating the following financial disclosure:

 [The funders had no role in study design, data collection and analysis, decision to

publish, or preparation of the manuscript].

Please provide an amended Funding Statement that declares *all* the funding or sources of support received during this specific study (whether external or internal to your organization) as detailed online in our guide for authors at http://journals.plos.org/plosone/s/submit-now.  Please state what role the funders took in the study.  If any authors received a salary from any of your funders, please state which authors and which funder. If the funders had no role, please state: "The funders had no role in study design, data collection and analysis, decision to publish, or preparation of the manuscript."

Reviewers' comments:

Reviewer's Responses to Questions

**Comments to the Author**

1. Is the manuscript technically sound, and do the data support the conclusions?

Reviewer #1: Yes

Reviewer #2: Yes

2. Has the statistical analysis been performed appropriately and rigorously? 

Reviewer #1: Yes

Reviewer #2: Yes

3. Have the authors made all data underlying the findings in their manuscript fully available?

Reviewer #1: Yes

Reviewer #2: Yes

4. Is the manuscript presented in an intelligible fashion and written in standard English?

Reviewer #1: Yes

Reviewer #2: No

5. Review Comments to the Author

Reviewer #1: In this study the authors report elevated levels of neuron-specific enolase (NSE) in CSF following birth asphyxia, and correlate the levels to grade of encephalopathy, aEEG and MRI findings, as well as cognitive and motor outcome. NSE as a marker for brain injury and adverse outcome following birth asphyxia has previously been widely studied in serum as well as CSF, including studies that report elevated NSE in CSF in association with adverse outcome in cooled infants. The strength of this study, is that NSE is studied in a relatively large group of infants treated according to current protocols for therapeutic hypothermia, that a wider variety of clinical variables indicating significant brain injury are studied, and that the authors present cut-off values that are highly predictive for brain injury and adverse outcome. Introduction and Discussion are succinct and to the point. Patient material is clearly defined and clinical examinations and definitions widely accepted.

There are, however, issues mainly regarding presentation and analysis of data that need to be addressed

Major comments:

Methods:

One of the main motivations for this study is to investigate NSE as a marker in the age of hypothermia. Three infants with severe HIE were not cooled according to clinical protocol but still included. Reasons for inclusion?

Statistics and presentation:

- Only Mann-Whitney is stated as method for comparisons between groups (numerical values for NSE between groups). In Table 1, control infants are compared with asphyxiated infants with three different grades of encephalopathy. Both numerical and nominal variables are compared and only one p-value per variable is presented. Statistics? Kruskall-Wallis/Chi-Square – or controls compared to all HIE groups together? Add to statistics under Methods.

- Statistical method needs to be added to all Figure and Table legends to make them self-explanatory.

- Comparing NSE levels between controls and asphyxiated infants with different grades of HIE (line 265-271) seem to be performed with multiple Mann-Whitney tests without correction. Please clarify/motivate or use other method including corrections

- Use of regression analyses for confounding factors – last part of Results. It is highly relevant and important to correct for age at lumbar puncture. Is the correction performed within the asphyxia group only, or in all infants including controls?

- In the same section of results: What is the rational for using regression analysis to correct for severity of HIE as it is strongly correlated with NSE levels as well as outcome? If I misunderstood this part, please clarify.

Pictures and legends:

- Legends for Pictures 2 and 3 have obviously been mixed up!

- Add number of infants (n) in the different groups into figures or legends

Discussion:

The authors report a remarkably high sensitivity and specificity for CSF NSE as a marker for neurological signs and brain injury as well as outcome. This a major finding that warrants further emphasis in the Discussion.

Minor comments:

- Please use consistent terminology. PCI is introduced instead of CP without explanation.

- Timing and comparisons of MRI. As I understand infants were either imaged twice (early around day 4 and late during second week) or only early. If two images were obtained only the later was used. Why - when comparing with children with only early images? Motivate or clarify.

Reviewer #2: In this prospective observational cohort study Leon-Lozanzo et al examined NSE in the CSF of a cohort of neonates with HIE. A control group of term neonates without CNS infection who has an LP taken for an infection screen was used for comparison. Within the cohort of HIE neonates, the level of NSE was analysed for short and long term outcome measures. The results of this study suggest that CSF NSE taken in the first few days after birth in neonates with HIE is associated with HIE severity and outcome and might be used as a predictive tool. CSF NSE as a biomarker for HIE outcome has been reported previously, as noted in the manuscript. The novelty of this study, according to the authors, is the association of CSF NSE with other surrogate outcome biomarkers (EEG and MRI).

The study is appropriately designed, and the results support the conclusions drawn. The results are of interest to clinicians and scientists in the field. There are some major and minor issues with the manuscript which I have highlighted below.

1.0 Major issues:

1.1 I suggest the authors consider hiring a copy editor or similar. The English language requires improvement, there are a number of grammatical/typing errors and the manuscript could be less verbose.

1.2 The main question a reader might want answered by this manuscript is ‘does CSF NSE, in isolation or in combination with other biomarkers, provide greater predictive valve for long term outcome than the more readily available tools, i.e. clinical assessment of severity, EEG and MRI? i.e. is it worth performed an LP in a neonate with HIE?’. The discussion highlights the pitfalls in the predicting outcome using EEG and MRI but misses the opportunity to tell us if CSF NSE is better or additive. A further analysis of the available data could shed light on this important question

2.0 Minor issues:

2.1 Some terms are used in the manuscript that are not in common use i.e. eutocic (line 226 and table 1) or may have translated poorly, i.e. ‘the aggression’ (line 370, 374, 413). The authors should substitute for the more commonly used terms

2.2 Replace phentanyl with fentanyl

2.3 In the introduction (lines 84 – 86) and discussion (lines 401 – 403) mention is made of the technical factors which make serum NSE unreliable, including processing of the sample and the effect of temperature. It would be worthwhile including whether it is known if these same factors influence CSF samples.

2.4 In limitations, I suggest reference to the use of infants with suspected infection as control subjects

2.5 Clinical and electrographic seizures are grouped as an outcome measure. I make a note of this due to the poor correlation between ‘clinical’ and electrographic seizures, and the uncertainty around the interpretation of clinical seizures without electrographic correlate. Consider including only electrographic seizures in analysis, or if the combined group is used, then the method of diagnosing clinical seizures needs to be added to the methods

2.6 Report IQR for age at ND assessment (line 180)

2.7 Page line 246, within description for Table 1. Footnote for ‘c’ incorrect (repeated from ‘b’)

2.8 Define PCI in full at least once (page line 261 and 321)

2.9 Figure 1. 2 outliers not shown on the figure (2855 ng/ml and 1490 ng/ml for 2 neonates with severe HIE referred to elsewhere in manuscript). Suggest either include in the figure as outlier dots or mention omission in the figure legend.

2.10 Descriptions for figure 2 and figure 3 need to be swapped around

2.11 Figure 2 and Table 2 include identical data. Both are not necessary

2.12 Consider showing significant differences between groups on figure 1 and figure 2

2.13 Line 54 and 318 refer to ‘minor findings’. Define this or replace with clearer language

2.14 Figure 3 – this is a very worthwhile figure, but the legend requires more detail. Explain the different shapes, the meaning of empty versus filled shapes more clearly, and the reason for the dashed line

2.15 The paragraph about ‘window of opportunity’, line 373 to 380, would benefit from more delicate wording or a clearer explanation (i.e. line 378 add… survive with subjectively intolerable disability)

6. PLOS authors have the option to publish the peer review history of their article (what does this mean?). If published, this will include your full peer review and any attached files.

Reviewer #1: No

Reviewer #2: No

---

## [Author Response · Author response to Decision Letter 0]

17 Apr 2020

PONE-D-19-29027

Cerebrospinal fluid levels of neuron-specific enolase predict the severity of brain damage in newborns with neonatal hypoxic-ischemic encephalopathy treated with hypothermia

PLOS ONE

Dear Dr Garcia-Alix,

Thank you for submitting your manuscript to PLOS ONE. After careful consideration, we feel that it has merit but does not fully meet PLOS ONE’s publication criteria as it currently stands. Therefore, we invite you to submit a revised version of the manuscript that addresses the points raised during the review process.

Please address the comments raised by both Reviewers, especially the major comments in order to further improve your manuscript.

We would appreciate receiving your revised manuscript by Apr 17 2020 11:59PM. To enhance the reproducibility of your results, we recommend that if applicable you deposit your laboratory protocols in protocols.io, where a protocol can be assigned its own identifier (DOI) such that it can be cited independently in the future. For instructions see:http://journals.plos.org/plosone/s/submission-guidelines#loc-laboratory-protocols

• A rebuttal letter that responds to each point raised by the academic editor and reviewer(s). This letter should be uploaded as separate file and labeled 'Response to Reviewers'.

• A marked-up copy of your manuscript that highlights changes made to the original version. This file should be uploaded as separate file and labeled 'Revised Manuscript with Track Changes'.

• An unmarked version of your revised paper without tracked changes. This file should be uploaded as separate file and labeled 'Manuscript'.

We look forward to receiving your revised manuscript.

Kind regards,

Pierre Gressens

Academic Editor

PLOS ONE

Journal Requirements:

Thank you. Please note that all our answers are referred to line numbers in the marked-up-copy that highlights changes made to the original version 

http://www.journals.plos.org/plosone/s/file?id=wjVg/PLOSOne_formatting_sample_main_body.pdfandhttp://www.journals.plos.org/plosone/s/file?id=ba62/PLOSOne_formatting_sample_title_authors_affiliations.pdf

Response. Thank you. We have checked our manuscript to insure that we meet PLOS ONE´s requirements.

2) Please include captions for your Supporting Information files at the end of your manuscript, and update any in-text citations to match accordingly. Please see our Supporting Information guidelines for more information: http://journals.plos.org/plosone/s/supporting-information.

Response. Thank you. We added captions for our supporting information files at the end of our manuscript and updated text citations to match accordingly. Please see line 839 at the end of the manuscript, and line 562 for in-text citations

3) We noticed you have some minor occurrence of overlapping text with the following previous publication(s), which needs to be addressed:

https://www.pedneur.com/article/S0887-8994(18)31326-2/fulltext

The text that needs to be addressed is in the Discussion section.

In your revision ensure you cite all your sources (including your own works), and quote or rephrase any duplicated text outside the methods section. Further consideration is dependent on these concerns being addressed.

Response. Thank you. We have reviewed the overlapping text and have rephrased any duplicated text outside the methods section. Please see text from line 418 to line 467 in the discussion section.

4) Please change your reference to "p=0.000" to "p<0.001" or as similarly appropriate, as p values cannot equal zero.

Response. Thank you. We have changed p=0.000 to p<0.001 in Tables 1 and 2

5) Thank you for your ethics statement: "Written information was given to the parents and written consent was obtained at the bedside on admission of each infant after explanation of the study and before its onset. The Human Studies Committee of each institution approved the study"

For additional information about PLOS ONE ethical requirements for human subjects research, please refer tohttp://journals.plos.org/plosone/s/submission-guidelines#loc-human-subjects-research.

Response. Thank you. We have completed the information in the ethics statement. Please see line 230.

6) We note you have included a table to which you do not refer in the text of your manuscript. Please ensure that you refer to Table 3 in your text; if accepted, production will need this reference to link the reader to the Table.

Response. Thank you. We have added the reference to Table 3 in the text (at the end of the paragraph about CSF-NSE and outcome in the results section).

7) Thank you for stating the following financial disclosure:

[The funders had no role in study design, data collection and analysis, decision to

publish, or preparation of the manuscript].

a. Please provide an amended Funding Statement that declares *all* the funding or sources of support received during this specific study (whether external or internal to your organization) as detailed online in our guide for authors athttp://journals.plos.org/plosone/s/submit-now.

b. Please state what role the funders took in the study. If any authors received a salary from any of your funders, please state which authors and which funder. If the funders had no role, please state: "The funders had no role in study design, data collection and analysis, decision to publish, or preparation of the manuscript." Please include your amended statements within your cover letter; we will change the online submission form on your behalf.

Statement received by email (April 16):

Please clarify the sources of funding (financial or material support) for your study. List the grants or organizations that supported your study, including funding received from your institution. 

Response. Thank you. We have added the following statement to the cover letter: “The authors received no specific funding for this work”.

Reviewers' comments:

Reviewer's Responses to Questions

Comments to the Author

1. Is the manuscript technically sound, and do the data support the conclusions?

Reviewer #1: Yes

Reviewer #2: Yes

2. Has the statistical analysis been performed appropriately and rigorously? 

Reviewer #1: Yes

Reviewer #2: Yes

3. Have the authors made all data underlying the findings in their manuscript fully available?

Reviewer #1: Yes

Reviewer #2: Yes

4. Is the manuscript presented in an intelligible fashion and written in standard English?

Reviewer #1: Yes

Reviewer #2: No

5. Review Comments to the Author

Reviewer #1: In this study the authors report elevated levels of neuron-specific enolase (NSE) in CSF following birth asphyxia, and correlate the levels to grade of encephalopathy, aEEG and MRI findings, as well as cognitive and motor outcome. NSE as a marker for brain injury and adverse outcome following birth asphyxia has previously been widely studied in serum as well as CSF, including studies that report elevated NSE in CSF in association with adverse outcome in cooled infants. The strength of this study, is that NSE is studied in a relatively large group of infants treated according to current protocols for therapeutic hypothermia, that a wider variety of clinical variables indicating significant brain injury are studied, and that the authors present cut-off values that are highly predictive for brain injury and adverse outcome. Introduction and Discussion are succinct and to the point. Patient material is clearly defined and clinical examinations and definitions widely accepted.

There are, however, issues mainly regarding presentation and analysis of data that need to be addressed

Major comments:

Methods:

One of the main motivations for this study is to investigate NSE as a marker in the age of hypothermia. Three infants with severe HIE were not cooled according to clinical protocol but still included. Reasons for inclusion?

Response. As the reviewer says, our intention was to describe the usefulness of NSE at the present time when therapeutic hypothermia is a standardized treatment in most countries. However, it is known that some patients do not receive this treatment for different reasons even though they were candidates. That is why we believe it is relevant to provide real data concerning why it is that some infants do not receive hypothermia. As stated in the text, in our series 3 newborns with severe HIE were not cooled: 166, 281, and 2855 ng/ml of CSF-NSE values. 

However, as the reviewer notes, including these three children could influence the results if we wanted to draw conclusions regarding NSE in cooled infants. Thus, we have repeated the analyses without including these 3 children. The results are very similar. We decided to add a comment in the discussion section (line 559).

We added the analysis eliminating the three uncooled infants as supplementary material, in case the reader is interested in having a look at them (S1 and S2 Tables). If the editor and/or reviewers consider that these tables do not need to be added as supplementary material, we agree to their removal.

Statistics and presentation:

- Only Mann-Whitney is stated as method for comparisons between groups (numerical values for NSE between groups). In Table 1, control infants are compared with asphyxiated infants with three different grades of encephalopathy. Both numerical and nominal variables are compared and only one p-value per variable is presented. Statistics? Kruskall-Wallis/Chi-Square – or controls compared to all HIE groups together? Add to statistics under Methods.

Response. Thank you. In Table 1, comparisons are made between different grades of encephalopathy, but controls are not included. That is why the paragraph in the text (lines 247 to 254) does not refer to Table 1. Table 1 is referred to in the text only when addressing the characteristics of the infants with HIE. Thus, p values in Table 1 only refer to the comparison of the different grades of encephalopathy. We have clarified this in the legend of Table 1 (lines 258 to 260) and in the methods section (lines 194 to 198).

- Statistical method needs to be added to all Figure and Table legends to make them self-explanatory.

Response. Thank you. We have added the appropriate information to all figures and tables. 

- Comparing NSE levels between controls and asphyxiated infants with different grades of HIE (line 265-271) seem to be performed with multiple Mann-Whitney tests without correction. Please clarify/motivate or use other method including corrections

Response. Thank you. This is a very appropriate suggestion. Before using the method with corrections we explored data trends and we noticed that an ANOVA post hoc analysis could better explore the data than using p-adjustment by Bonferroni. So, we transformed the CSF-NSE variable to normalize it (logarithm transformation) and performed a parametric test, which allowed analysis between groups without losing power. As sample size of the moderate and severe group was not large, significance may be affected using a non-parametric test. 

The analysis showed that there were statistically significant differences in all comparisons between groups. Same results were found when we excluded the three infants that were not cooled. We added the analysis as supplementary information in case it wants to be checked (but we did not included it in the manuscript as it is too long) and a paragraph referring to this correction in the methods section (line 199).

- Use of regression analyses for confounding factors – last part of Results. It is highly relevant and important to correct for age at lumbar puncture. Is the correction performed within the asphyxia group only, or in all infants including controls?

Response. As the reviewer says, regression analysis to correct CSF-NSE levels for age at lumbar puncture was performed in all infants including controls. The analysis was performed by including time as a numerical variable, and also categorizing it into two periods <48 and > 48 hours. In both cases, time was not a confounding factor.

To better display the information we have changed the sentence on line 394 and moved the information regarding the regression analysis focused on the degree of HIE to line 294. We have left the regression analysis with respect to the outcome more clearly reported in lines 394 to 396 . 

We also changed the paragraph in the methods section in order to make it more comprehensible (line 206)

- In the same section of results: What is the rational for using regression analysis to correct for severity of HIE as it is strongly correlated with NSE levels as well as outcome? If I misunderstood this part, please clarify.

Response. Thank you. We checked this issue with our statistics department and they agree that in order to estimate the predictive capacity of the NSE regardless of the degree of HIE and the time of determination, the NSE-outcome association should be estimated by multiple regression, adjusting for these covariates. This adjusted estimation helps to distinguish the outcome differences associated with NSE from those associated with the degree of HIE (the effect of NSE levels may be different depending on the degree of HIE).

In any case, as we noted above, we changed the paragraph at the end of the results section to explain the information more clearly (lines 206 to 211).

Pictures and legends:

- Legends for Pictures 2 and 3 have obviously been mixed up!

Response. Yes, thank you. We have corrected the legends.

- Add number of infants (n) in the different groups into figures or legends

Response. We completed legends for Figures. Numbers were already added in the tables.

Discussion:

The authors report a remarkably high sensitivity and specificity for CSF NSE as a marker for neurological signs and brain injury as well as outcome. This a major finding that warrants further emphasis in the Discussion.

Response. Thank you. To give more emphasis we moved the last paragraph at the end of the discussion section (lines 569-574) and rephrased the information in lines 542 to 550.

Minor comments:

- Please use consistent terminology. PCI is introduced instead of CP without explanation.

Response. Thank you. We changed PCI to CP (line 358 and 280).

- Timing and comparisons of MRI. As I understand infants were either imaged twice (early around day 4 and late during second week) or only early. If two images were obtained only the later was used. Why - when comparing with children with only early images? Motivate or clarify.

Response. Thank you very much for your observation. We have eliminated the word “early” when referring to the infant who underwent only one study in order not to be misunderstood. For infants in whom only one MRI study was performed, the time ranged from 38 to 584 days of life. That is why we preferred to use the “late MRI” in those with two images, in order to indicate as clearly as we could the severity of the damage. We know this could be a controversial aspect as an early MRI could be comparable to a late MRI and may even offer some advantage, especially if sequences such as diffusion and spectroscopy are incorporated. 

In any case, we also explored the analyses using the early image rather than late MRI, and the results were very similar (without differences when excluding the three non-cooled infants) with slightly better area under the curve in the ROC analysis when using late MRI.

Variable N Cutoff-point Area under the ROC (95%CI) Sensitivity (95%CI) Specificity (95%CI) PPV (95%CI) NPV (95%CI)

Using the late MRI if two images were obtained 

Moderate-severe injury (MRI) 20 80 0.93 (0.85,1.00) 0.75 (0.51,0.91) 1.00 (0.81,-) 1.00 (0.79,1.00) 0.78 (0.55,-)

Global injury pattern (MRI) 17 80 0.95 (0.88,1.00) 0.82 (0.57,0.96) 0.95 (0.76,0.99) 0.93 (0.69,0.99) 0.87 (0.65,0.99)

Using the early MRI if two images were obtained 

Moderate-severe injury (MRI) 21 80 0.89 (0.80,0.99) 0.71 (0.48,0.89) 1.00 (0.80,-) 1.00 (0.79,1.00) 0.74 (0.51,-)

Global injury pattern (MRI) 23 80 0.87 (0.77,0.98) 0.65 (0.43,0.84) 1.00 (0.78,-) 1.00 (0.78,1.00) 0.65 (0.43,-)

Reviewer #2: In this prospective observational cohort study Leon-Lozanzo et al examined NSE in the CSF of a cohort of neonates with HIE. A control group of term neonates without CNS infection who has an LP taken for an infection screen was used for comparison. Within the cohort of HIE neonates, the level of NSE was analysed for short and long term outcome measures. The results of this study suggest that CSF NSE taken in the first few days after birth in neonates with HIE is associated with HIE severity and outcome and might be used as a predictive tool. CSF NSE as a biomarker for HIE outcome has been reported previously, as noted in the manuscript. The novelty of this study, according to the authors, is the association of CSF NSE with other surrogate outcome biomarkers (EEG and MRI).

The study is appropriately designed, and the results support the conclusions drawn. The results are of interest to clinicians and scientists in the field. There are some major and minor issues with the manuscript which I have highlighted below.

1.0 Major issues:

1.1 I suggest the authors consider hiring a copy editor or similar. The English language requires improvement, there are a number of grammatical/typing errors and the manuscript could be less verbose.

Response. Thank you. We have reviewed the language and changed it where appropriate.

1.2 The main question a reader might want answered by this manuscript is ‘does CSF NSE, in isolation or in combination with other biomarkers, provide greater predictive valve for long term outcome than the more readily available tools, i.e. clinical assessment of severity, EEG and MRI? i.e. is it worth performed an LP in a neonate with HIE?’. The discussion highlights the pitfalls in the predicting outcome using EEG and MRI but misses the opportunity to tell us if CSF NSE is better or additive. A further analysis of the available data could shed light on this important question

Response. We thank the reviewer for this comment as it demonstrates his/her clinical interest is in line with ours. In our opinion CSF-NSE is one more pillar when it comes to establishing the severity of the damage and relating it to the prognosis. As our research points out, it reflects the damage revealed by other tools such as clinical exam, aEEG, and neuroimaging. In our opinion, CSF-NSE should be considered as a key tool when it comes to obtaining as much information as possible regarding each infant. However, although we understand what the reviewer points out and its relevance, performing a combined analysis with all the tools would be advisable with a greater number of patients included for better consistency of statistical models. We are working on acquiring this data set shortly. 

But to help the readers compare the probability values for each tool, we added a paragraph (line 397) in the results section and also Table 4 (line 400) in which it may be seen that the CSF-NSE compares favorably with the other classically used tools. We also added an appropriate sentence in the discussion section (line 538).

2.0 Minor issues:

2.1 Some terms are used in the manuscript that are not in common use i.e. eutocic (line 226 and table 1) or may have translated poorly, i.e. ‘the aggression’ (line 370, 374, 413). The authors should substitute for the more commonly used terms

Response. Thank you. We changed these two terms: we replaced ‘eutocic’ delivery with ‘spontaneous’ and hypoxic-ischemic ‘aggression’ with hypoxic-ischemic ‘insult’. 

2.2 Replace phentanyl with fentanyl

Response. Thank you. We have corrected this.

2.3 In the introduction (lines 84 – 86) and discussion (lines 401 – 403) mention is made of the technical factors which make serum NSE unreliable, including processing of the sample and the effect of temperature. It would be worthwhile including whether it is known if these same factors influence CSF samples.

Response. Thank you very much for pointing out our imprecision on this issue. We have completed the phrase in the introduction (lines 8e to 88), also in the methods section (line 136), and we have tried to better clarify in the discussion (line 484) with a new reference (number 53). We also added a comment regarding the measuring method that may also be relevant in addressing the confounding technical factors (lines 486 to 488).

2.4 In limitations, I suggest reference to the use of infants with suspected infection as control subjects

Response. Thank you. We have completed the information and added references already included in the manuscript (numbers 21,26 and 27) of other studies that used the same approach as ours (lines 563 to 568).

2.5 Clinical and electrographic seizures are grouped as an outcome measure. I make a note of this due to the poor correlation between ‘clinical’ and electrographic seizures, and the uncertainty around the interpretation of clinical seizures without electrographic correlate. Consider including only electrographic seizures in analysis, or if the combined group is used, then the method of diagnosing clinical seizures needs to be added to the methods

Response. Thank you. We agree with the reviewer that it may be better to consider only electrographic seizures in the analysis. Thus, we re-analyzed this variable and modified Tables 1, 2, and 3. At any rate, the results were very similar to those of previous analyses.

2.6 Report IQR for age at ND assessment (line 180)

Response. Thank you. We have added the information.

2.7 Page line 246, within description for Table 1. Footnote for ‘c’ incorrect (repeated from ‘b’)

Response. Thank you. We have corrected it.

2.8 Define PCI in full at least once (page line 261 and 321)

Response. Thank you. This is a mistake. PCI is the abbreviation for cerebral palsy in Spanish. We have changed it to the correct abbreviation in English: CP. 

2.9 Figure 1. 2 outliers not shown on the figure (2855 ng/ml and 1490 ng/ml for 2 neonates with severe HIE referred to elsewhere in manuscript). Suggest either include in the figure as outlier dots or mention omission in the figure legend.

Response. Since Figures 1 and 2 represent the enolase values in medians and interquartile range, those values outside P25 and P75 are not shown. If they were to be displayed, it would not have to be just the two values that the reviewer refers to, but all of them. In our opinion, this makes the graph ‘dirtier’; it is common practice to show only the median and the interquartile range when drawing the data in box-plots. Figure 3 does point out that these two patients are not represented as it is a figure in which all patients in the cohort are represented and is therefore not a half-IQR as Figures 1 and 2.

2.10 Descriptions for figure 2 and figure 3 need to be swapped around

Response. Thank you. We have corrected it.

2.11 Figure 2 and Table 2 include identical data. Both are not necessary

Response. We have eliminated Figure 2 and renamed the Fig numbers: Fig 3 has been changed to Fig 2

2.12 Consider showing significant differences between groups on figure 1 and figure 2

Response. Relevant information was added to the legend of Figure 1. Figure 2 was eliminated.

2.13 Line 54 and 318 refer to ‘minor findings’. Define this or replace with clearer language

Response. The reason for pointing out normal or minor findings was to refer to the outcome that did not correspond to "adverse outcome". But as the reviewer says, this could be misunderstood, so we've removed it from lines 55 and 355.

2.14 Figure 3 – this is a very worthwhile figure, but the legend requires more detail. Explain the different shapes, the meaning of empty versus filled shapes more clearly, and the reason for the dashed line

Response. Thank you. We have completed and clarified the information. Please note that Figure 3 changed the name to Figure 2.

2.15 The paragraph about ‘window of opportunity’, line 373 to 380, would benefit from more delicate wording or a clearer explanation (i.e. line 378 add… survive with subjectively intolerable disability)

Response. Thank you. We have added “with subjectively intolerable disability”.

6. PLOS authors have the option to publish the peer review history of their article (what does this mean?). If published, this will include your full peer review and any attached files.

Do you want your identity to be public for this peer review? For information about this choice, including consent withdrawal, please see our Privacy Policy.

Reviewer #1: No

Reviewer #2: No

---

## [Decision Letter · Decision Letter 1]

19 May 2020

Cerebrospinal fluid levels of neuron-specific enolase predict the severity of brain damage in newborns with neonatal hypoxic-ischemic encephalopathy treated with hypothermia

PONE-D-19-29027R1

Dear Dr. Garcia-Alix,

We are pleased to inform you that your manuscript has been judged scientifically suitable for publication and will be formally accepted for publication once it complies with all outstanding technical requirements.

With kind regards,

Pierre Gressens

Academic Editor

PLOS ONE

Additional Editor Comments (optional):

Reviewers' comments:

Reviewer's Responses to Questions

**Comments to the Author**

1. If the authors have adequately addressed your comments raised in a previous round of review and you feel that this manuscript is now acceptable for publication, you may indicate that here to bypass the “Comments to the Author” section, enter your conflict of interest statement in the “Confidential to Editor” section, and submit your "Accept" recommendation.

Reviewer #2: All comments have been addressed

2. Is the manuscript technically sound, and do the data support the conclusions?

Reviewer #2: Yes

3. Has the statistical analysis been performed appropriately and rigorously? 

Reviewer #2: Yes

4. Have the authors made all data underlying the findings in their manuscript fully available?

Reviewer #2: Yes

5. Is the manuscript presented in an intelligible fashion and written in standard English?

Reviewer #2: Yes

6. Review Comments to the Author

Reviewer #2: The authors have satisfactorily addressed the comments and suggestions made. The manuscript is improved by the changes made. I have no further comments.

7. PLOS authors have the option to publish the peer review history of their article (what does this mean?). If published, this will include your full peer review and any attached files.

Reviewer #2: No

---

## [Editor Report · Acceptance letter]

22 May 2020

PONE-D-19-29027R1 

Cerebrospinal fluid levels of neuron-specific enolase predict the severity of brain damage in newborns with neonatal hypoxic-ischemic encephalopathy treated with hypothermia 

Dear Dr. Garcia-Alix:

I am pleased to inform you that your manuscript has been deemed suitable for publication in PLOS ONE. Congratulations! Your manuscript is now with our production department. 

With kind regards,

on behalf of

Dr. Pierre Gressens 

Academic Editor

PLOS ONE